# The Derivative Difluoroboranyl-Fluoroquinolone “7a” Generates Effective Inhibition Against the *S. aureus* Strain in a Murine Model of Acute Pneumonia

**DOI:** 10.3390/cimb47020110

**Published:** 2025-02-10

**Authors:** L. Angel Veyna-Hurtado, Hiram Hernández-López, Fuensanta del Rocío Reyes-Escobedo, Denisse de Loera, Salvador García-Cruz, Lorena Troncoso-Vázquez, Marisol Galván-Valencia, Julio E. Castañeda-Delgado, Alberto Rafael Cervantes-Villagrana

**Affiliations:** 1Doctorado en Ciencias Farmacobiológicas, Facultad de Ciencias Químicas, Universidad Autónoma de San Luis Potosí, San Luis Potosí 78210, Mexico; angelveyna@gmail.com; 2Laboratorio de Investigación en Síntesis Orgánica y Modificación Química, Unidad Académica de Ciencias Químicas, Universidad Autónoma de Zacatecas, Zacatecas 98160, Mexico; hiram_hdez@hotmail.com; 3Laboratorio de Microbiología Sanitaria de Investigación y Servicio al Público, Unidad Académica de Ciencias Químicas, Universidad Autónoma de Zacatecas, Zacatecas 98160, Mexico; fuenreyes@uaz.edu.mx; 4Facultad de Ciencias Químicas, Universidad Autónoma de San Luis Potosí, San Luis Potosí 78210, Mexico; atenea.deloera@uaslp.mx; 5Laboratorio de Cirugía, Unidad Académica de Medicina Humana, Universidad Autónoma de Zacatecas, Zacatecas 98160, Mexico; sgc_54@uaz.edu.mx; 6Departamento de Patología, Instituto Mexicano del Seguro Social (IMSS), Zacatecas 98160, Mexico; lorecrm@hotmail.com; 7Laboratorio de Investigación en Patología y Productos Naturales, Unidad Académica de Ciencias Químicas, Universidad Autónoma de Zacatecas, Zacatecas 98160, Mexico; gavm001144@uaz.edu.mx; 8Investigador por México/Cátedras CONAHCYT (SECIHTI), Unidad de Investigación Biomédica de Zacatecas, Instituto Mexicano del Seguro Social (IMSS), Zacatecas 98160, Mexico; julioenriquecastaneda@gmail.com; 9Laboratorio de Investigación en Terapéutica Experimental, Unidad Académica de Ciencias Químicas, Universidad Autónoma de Zacatecas, Zacatecas 98160, Mexico

**Keywords:** fluoroquinolone, *S. aureus*, antimicrobial, molecular docking, pneumonia, histopathology

## Abstract

During the last decades, most bacterial strains have become increasingly resistant to antibiotics. This led the WHO to declare a global emergency in 2017 and urge the development of new active compounds. Some families of antibiotics still show high antibacterial efficacy, as is the case of fluoroquinolones, which have a broad spectrum of action. For this reason, our research group derived several compounds from fluoroquinolones, selecting a compound with good antibacterial activity for further evaluations, a difluoroboranil-fluoroquinolone complex labeled **7a**. Antibacterial activity was evaluated using the Kirby–Bauer method against *S. aureus* (clinical isolate HGZ2201#ID). The MIC and MBC were obtained by macrodilutions and reseeding. In vivo antimicrobial activity was evaluated in a Balb/c mouse model infected intratracheally with *S. aureus* and subsequently treated with ciprofloxacin or **7a** (80 mg/kg/day) for five days. A mean of 8.55 ± 0.395 cm^2^ inhibition area was observed using **7a**, while ciprofloxacin generated a mean inhibition of 7.86 ± 0.231 cm^2^. Compound **7a** showed a MIC and MBC of 0.25 μg/mL. This reduced the generation of pneumonic lung tissue to 5.83%, while the untreated infected group generated 60.51% of pneumonic tissue. Compound **7a** proved to be an antimicrobial agent capable of inhibiting the in vitro development of *S. aureus*. Furthermore, **7a** showed effectiveness in decreasing the progression of acute pneumonia induced by *S. aureus* in a murine model.

## 1. Introduction

The WHO has published reports stating that the large increase in drug resistance could lead to a post-antibiotic era by 2050. It has also been established that by then there will be 10 million deaths per year from antibiotic-resistant infections [1,2,3]. One of the most problematic bacteria-related diseases, acute respiratory infections (ARI), cause millions of deaths worldwide, representing the first leading cause of death in children and the sixth leading cause of death in adults [4,5,6,7,8]. *Staphylococcus aureus* is one of the main causes of food poisoning in industrialized countries [9], and a causal agent of pneumonia due to its infectious capacity. Regarding Community-Acquired Pneumonia (CAP), about 5–10% of patients end up in the intensive care unit. Severe CAP represents the most complicated subgroup to treat in this pathology, whose mortality can reach 40% in patients with septic shock who require mechanical ventilation [10,11,12].

In a global review about mortality from bacterial infections, it is reported that *S. aureus* was the leading bacterial cause of death in 135 countries (including Mexico and the USA), and was also associated with most deaths in individuals over 15 years of age, worldwide. In 2019, more than 6 million deaths resulted from three bacterial infectious syndromes, of which lower respiratory infections and bloodstream infections each caused more than 2 million deaths [13].

The fluoroquinolone family have an extended use in clinics for complicated infections, and the recent derivatized structures show high activity and a broad antibacterial spectrum, being effective against *S. aureus*, *K. pneumoniae*, and *E. coli*, even presenting a high utility against difficult treatment bacteria, such as *Y. enterocolitica, Mycobacteria, Salmonella, Shigella, Campylobacter* [14,15]. These fluoroquinolones have evidenced a decreased bacterial resistance, phototoxicity has been eliminated, and they are effective in methicillin-resistant *S. aureus* strains [16,17]. Previously, our research group have synthesized and characterized several fluoroquinolone compounds in a complex with boron atoms [18] in order to increase the antibacterial activity and improve the physicochemical and pharmacokinetic properties [19].

In the organic synthesis of the compounds, the addition of functional groups that improve the overall potency were considered, such as the addition of cyclopropyl in position N-1, also achieved by the addition of an ethyl group [20]. In turn, a fluorine group was added at position 6, which has traditionally been reported to increase the activity against Gram-negative bacteria. This activity is also enhanced by the piperazine at the C-7 position of the quinolone ring, compared to other compounds lacking such a structure [20,21,22].

Boron–fluoroquinolone derivatives are obtained by incorporating the boron atom at positions 3 and 4 through keto and carboxylic groups. Currently, there are few reports of boronated fluoroquinolones, with a small number of examples with inhibitory activity in cancer cell lines or bacteria [19,23,24,25]. So, it is required to expand the libraries of boron derivatives and their conjugates, and elucidate their mechanism of action. In addition, molecular docking studies have been described using norfloxacin–boron complexes as ligands, resulting in an increase in the interaction obtained when boron was bound to fluoride atoms [26].

Boron has unique chemical properties, such as its ability to form stable complexes with biomolecules, which can enhance antibacterial activity, modulate target specificity, and reduce the cytotoxicity of some organic structure moieties. It has also been reported to have antibacterial activity on its own, boosting the immune system, having activity in vaginal infections, and has even been described to be useful in cancer treatment (boron neutron capture therapy [BNCT]) by minimizing the number of healthy cells affected [27,28,29,30]. However, there are few reports of fluoroquinolone derivatives containing boron-based substituents reported.

An in vitro screening revealed the compound difluoroboranyl 1-ethyl-6-fluoro-4-oxo-7-piperazin-1-il-1,4-dihydro-quinoline-3-carboxylate, later labeled “**7a**” (showed in Figure 1a) is one of the best inhibitors against *S. aureus* and *K. pneumoniae*. Our research team has already exhaustively evaluated **7a** in the Gram-negative strain *Klebsiella pneumoniae*, demonstrating a notorious antibacterial activity [31]. Therefore, our main objective in this research was to evaluate the antimicrobial effect in vitro of **7a** against Gram-positive bacteria strain *S. aureus*, and determine its activity in an acute pneumonia model in mice (the experimental design is shown in Figure 1b).

## 2. Materials and Methods

### 2.1. Synthesis of Difluoroboranyl 1-ethyl-7-fluoro-4-oxo-7-piperazin-1-il-1,4-dihydro-quinoline-3-carboxylate, ***7a***

The compound **7a** derived from fluoroquinolone, which, by forming a complex with boron, has an increased the biological activity [31], was synthesized according to Hernández-López et al.’s method [19], with some modifications. In a reflux system, we added 1.5 mL of acetonitrile, 69.4 μL (0.5 mmol) of triethylamine (TEA), 100 mg (332.85 μmol) of difluoroboryl 1-ethyl-6,7-difluoro-4-oxo-1,4-dihydro-quinoline-3-carboxylate, and 43 mg (0.5 mmol) of piperazine at 80 °C for 10 h with constant stirring. After that, 1 mL of ethanol was added to the reaction mixture, resulting in a light-yellow solid that was separated by vacuum filtration and washed with ethanol until a yellow solid was obtained, with a 79% reaction yield and a melting point (obtained by using a Fisher-Johns melting point apparatus) of 230–231 °C. The spectroscopy characterization of **7a** was in accordance with that reported [18]: ^1^H NMR (Varian Mercury plus 400 MHz spectrometer using TMS as the internal control, DMSO-*d*_6_) δ (ppm): 9.25 (*s*, 1H), 8.05 (*d*, *J*_HF orto_ = 13.48 Hz, 2H), 7.31 (*d*, *J*_HF meta_ = 7.31 Hz, 2H), 4.82 (*c*, *J*_HH_ = 7.11 Hz, 2H), 3.38 (m, 4H), 2.89 (m, 4H), 1.46 (t, *J*_HH_ = 7.11 Hz, 3H).

### 2.2. Molecular Docking

First, a molecular docking evaluation was carried out [32] in order to determine whether molecule **7a** would interact with DNA gyrase (the usual target of fluoroquinolones). We selected and downloaded the crystallized protein from the RSCB Protein Data Bank (PDB) with the ID 2xct, a 3.35 Å structure of *S. aureus* gyrase co-crystalized with ciprofloxacin.

The active sites of the protein were predicted in the Protein Plus server (https://proteins.plus/#dogsite, accessed on 5 February 2025). Structures adjacent to a single protein, such as the solvent, were removed with UCSF Chimera (1.16), except for the DNA double strand and the Mg^2+^ ion, which were retained. The same software allowed us to store the energy minimization of the protein, whose computations were performed with the PDB2PQR parameter set. We used the AutoDock Tools software (1.5.7) to give the receptor the protonation state.

The structure of **7a** was drafted in BIOBIA Draw software (19.1.0), and later opened in Avogadro (1.2.0) to conduct geometrical optimization with the UFF force field by using the steepest descent algorithm with 4 steps per update. AutoDock Tools (1.5.7) allowed us to establish the torsion tree of the ligand (molecule **7a**), and to design the grid center for use in the receptor. AutoDock Vina (1.1.2) was used to carry out the docking of **7a** against the gyrase protein. The post-dock results were analyzed and visualized using the Receptor–Ligand interactions on Discovery Studio Visualizer (21.1.0).

### 2.3. Kirby–Bauer Evaluation

The Kirby–Bauer technique was performed to evaluate the sensibility of the strain to the compound [33]. An isolate of *Methicillin Resistant Staphylococcus aureus* (MRSA) susceptible to fluoroquinolones, obtained from respiratory tract infections (HGZ2201#ID), was used for this experiment. Clinical isolates were obtained from the strain collection of the Universidad Autónoma de Zacatecas, where they were kept frozen at −80 °C until use. A 0.5 McFarland bacterial suspension of *S. aureus* (from the clinical isolate HGZ2201#ID) was inoculated on Mueller–Hinton agar plates (MH) (Bioxon, no. cat. 211667, Kowale, Poland). Disks impregnated with the **7a** (with 0.1, 1, 5, or 10 μg) diluted in DMSO were placed over the MH medium. Ciprofloxacin 5 μg (Ciproxin^®^, Bayer, Leverkusen, Germany) was used as a positive control, and dimethyl sulfoxide (DMSO) [10% concentration in aqueous dilution] (Sigma-Aldrich, cat. no. 472301, St. Louis, MO, USA) was used as a vehicle control. The inoculated growth medium plates were incubated for 24 h at 37 °C in an aerobic environment. Digital photographs of the Petri dishes were taken and analyzed with ImageJ (1.52) software with a reference metric scale, obtaining a pixel/mm ratio. Then, we used the “polygon” functionality to select the inhibition halo and finally select the “Analyze-Measure” options that allowed us to obtain the results in mm^2^ of inhibition.

### 2.4. Minimum Inhibitory Concentration

The Minimum Inhibitory Concentration (MIC) was determined by macrodilutions, using serial dilutions of the **7a** compound. We used between 32 μg/mL and 0.0625 μg/mL in Müller–Hinton nutrient broth (Bioxon, cat. no. 210300), and then a standard inoculum of 5 × 10^4^ CFU/mL of *S. aureus* was added. The tubes were then incubated for 24 h at 37 °C in aerobic medium, and bacterial growth was evidenced by opacity development. Tubes without visible turbidity were reseeded in MH medium to determine the Minimum Bactericidal Concentration (MBC) defined as the plate/concentration on which no CFU grew after incubation for 24 h at 37 °C.

### 2.5. Acute Pneumonia In Vivo Model

The murine acute pneumonia model consisted of sixteen male and female BALB/c albino mice between 10 and 12 weeks of age (age range in accordance with values reported in other models [34,35]). They were divided into 4 experimental groups with 4 mice per group, denominated as: control group without infection; *S. aureus* infection without treatment; *S. aureus* infection and administration of ciprofloxacin; *S. aureus* infection and administration of **7a**. All of them were housed in a pathogen-free environment with 12 h light/dark periods and received sterile food and water ad libitum. The animals were handled in accordance with the NOM-062-ZOO-1999 “technical specifications for the production, care, and use of laboratory animals” and the evaluations were accepted by the Institutional Ethics Committee of the Autonomous University of Zacatecas with acceptance code SACS/UAZ/308/2020. Mice were anesthetized using 1.5 mL/5 kg of sodium pentobarbital intraperitoneally (Cheminova, cat. no. 30375-B), and then inoculated with 50 μL of *S. aureus* bacterial suspension with 9 × 10^7^ CFU resuspended in distilled water through a 22-gauge cannula intratracheally, according to reported methods [36].

Compound **7a** was administered intraperitoneally 24 h after infection (route suitable for administration in small rodents) [36] with a dosage of 80 mg/kg/day during 5 consecutive days. The same treatment scheme was performed for ciprofloxacin, consistent with other models [37,38]. The mice’ body weights were measured daily [39]. The experimental animals were euthanized on the sixth day using a CO_2_ chamber. The lungs were perfused with 10% formalin by intratracheal inoculation, preserving their morphology [40], and then the right lung of each mouse was dissected. The lungs were embedded in paraffin and cut into 2 μm thick sections, Hematoxylin and Eosin (H&E) staining was subsequently performed. Microscopic histopathological analysis was performed on the sections at 40× using an Olympus^®^ inverted microscope [34,35,39]. Digital photographs of lung morphologies were taken and then analyzed using ImageJ software (1.52). This allowed us to establish a metric scale by obtaining a pixel/mm ratio using the “polygon” functionality, selecting the pneumonic areas/foci, identified as those in which the alveolar spaces were reduced and cellular infiltration in the lung tissue was evident. Then, the “Analyze-Measure” options allowed us to obtain results in cm2 to calculate the percentage of tissue area with pneumonia [41,42].

### 2.6. Statistical Analysis

The normality of the data was evaluated by the Shapiro–Wilk test. The data presented normal distribution and was analyzed with one-way ANOVA and Tukey’s post hoc test, values of *p* ≤ 0.05 were considered significant. Mean and standard deviation are used as descriptive statistics, and in the categorical concentration experiments, the mode was used. Graphs were performed in GraphPad Prism (8.0.2) software.

## 3. Results

As already established by the WHO, research and discovery into new antibiotic drugs must be a priority [43]. In this work, we evaluated the antimicrobial efficacy of **7a**, showing strong antimicrobial activity.

### 3.1. Compound ***7a*** Interacts In Silico with S. aureus DNA Gyrase

The molecular docking of **7a** with DNA gyrase of the *S. aureus* bacteria was performed, and the best position had the binding energy score of −10.1 kcal/mol. The interactions are represented in Figure 2a, where we can observe electrostatic and some hydrophobic interactions with different amino acids or nucleotides, showing a probable inhibition of the enzyme. We can observe hydrogen bonding and van der Waals interactions between **7a** and adenine (DA1368), guanine (DG1353), phenylalanine (PHE970), aspartate (ASP21), and glycine (GLY20 and GLY260), among others. In Figure 2b, we can observe that the amino acids in the interactions are presented close to the binding site of other fluoroquinolones, but they are not exactly the same. This is of therapeutic advantage since the mechanism of action is different, and this means that the use of this new class of antibiotics developed by our group could be used independently of the novel fluoroquinolones. We observed that molecule **7a** is oriented towards the upper part in the Gyr-A subunit of DNA gyrase and interactions of the fluorine groups in the boron complex with histidine (HIS259) and arginine (ARG969).

Moreover, the same methodology was applied to the structure of ciprofloxacin, finding a score of −8.9 kcal/mol, highlighting electrostatic interactions such as van der Waals and hydrogen bonds, with proline (PRO504) and asparagine (Asn447), respectively, among other interactions. Interactions with thymine (DT1340), adenine (DA1375), and guanine (DG1339) near the target site are also present, as can be seen in Figure 2c. Likewise, the most stable position of interaction of ciprofloxacin is located in the GyrA subunit, which is consistent with the preferred site of interaction for this drug in bacterial inhibition, as depicted in Figure 2d.

### 3.2. Compound ***7a*** Shows Antimicrobial Activity by Kirby–Bauer Assay

In order to provide experimental evidence of antimicrobial effects, **7a** was added to the *S. aureus* HGZ2201#ID strain plated on solid agar. The areas of inhibition in *S. aureus* were measured using Image J software (1.52) and graphically represented in Figure 3. After the statistical analysis, equivalence was observed in the use of ciprofloxacin and **7a** (using the same concentration of 5 μg), having obtained 8.549 ± 0.395 cm^2^ of mean inhibition in *S. aureus*. This result evidenced the activity of the compound, equivalent to the activity of the control ciprofloxacin that generated a mean inhibition of 7.86 ± 0.231 cm^2^, so we proposed to continue the evaluations of this quinolonic derivative. The other concentrations of **7a** also evidenced antibacterial activity, obtaining mean areas of inhibition of 1.86 cm^2^ with 0.1 μg (*p* < 0.0001, 95% CI: −0.16, 3.88); 4 cm^2^ with 1 μg (*p* < 0.0001, 95% CI: 3.04, 4.97); 8.55 cm^2^ with 5 μg (not significant, 95% CI: 7.57, 9.53); and 6.71 cm^2^ with 10 μg (not significant, 95% CI: 3.77, 9.66).

### 3.3. ***7a*** Antimicrobial Activity by MIC and MBC Shows Non-Inferiority to Ciprofloxacin

Regarding the MIC determination, ciprofloxacin was used as a control, resulting in 0.25 μg/mL against the strain of *S. aureus*, Subsequently, in the reseeding of the suspensions, the MBC resulted in 0.5 μg/mL. Derivatized **7a** obtained the same MIC of 0.25 μg/mL, matching its MBC of 0.25 μg/mL, as shown in Table 1. Compound **7a** exhibits a MIC characteristic of a compound active against *S. aureus* (≤1 μg/mL), according to CLSI guidelines [44].

### 3.4. Acute Pneumonia In Vivo Model

Subsequently, the in vivo acute pneumonia model was set. The bacterial suspension administered to the mice contained 9 × 10^7^ CFU of *S. aureus*, a close concentration to that reported by Esposito et al. (3 × 10^8^ CFU) [45], and by Kim and Missiakas (de 2–4 × 10^8^ CFU) [46]. The suspensions were optimal for our experimental development, as they allowed for the survival of biological reagents [42]. The suspension concentration in our model of pneumonia (higher than some reported concentrations), and the route of administration did not require an immunosuppressive treatment to generate infection in healthy mice. Other authors have used 100–250 mg of cyclophosphamide to establish a pneumonia model [47,48].

The in vivo model implemented was intended to be carried out without using other concomitant drugs such as cyclophosphamide, which is used to cause neutropenia in some pneumonia models, but that can diminish the classical signs of acute inflammation [42,48,49,50]. Therefore, we generated the infection with only the bacterial suspension, achieving a more homogeneous state among the experimental groups, as reported in the literature [34,51,52].

No significant change was observed in the daily weight of the groups. The uninfected group recorded increases of 0.5 g on day 4, and 1.16 g on day 7 of the experiment. In the untreated *S. aureus* infected group, a decrease of 1.56 g on average was recorded for day 4, and an increase of 1.93 g on day 7, due to the immune response [53]. Mice given ciprofloxacin tended to lose weight until the end of the experiment, without regaining it completely. This may be due to adverse reactions, such as nausea vomiting, diarrhea, abdominal pain, and hemolytic anemia, among others [54,55,56]. Contrary to this, mice infected and treated with **7a** showed weight patterns very similar to healthy mice, although no significant differences are observed in these measures between the groups.

The weight loss in mice models administered with ciprofloxacin was already reported by Zhu et al. [57] who suggested that this drug induced anorexia. The **7a** may lead to weight maintenance due to bactericidal capacity upon the infection, and it probably does not induce anorexia. Further evaluations are needed to detail the phenomenon of weight changes during the experimentation.

Once the lung tissues were obtained at the end of the in vivo evaluation, they were observed in their entirety, and eight digital image fields per mouse were taken and analyzed. The pneumonic percentage value in every lung was averaged per experimental group, with an n = four mice per group. A representative image of lung tissue is presented in Figure 4a,d, the blue arrowheads show the alveolar tissue without immune cell infiltration. Figure 4a shows the healthy lung tissue of mice without infection, with the pneumonic areas being significantly lower than the group with untreated *S. aureus*. These showed an average of 60.74% pneumonic area (indicated by the black arrowheads) as shown in Figure 4b (**** *p* < 0.0001, 95% CI: 57.10, 64.38). The significant decrease in the percentage of pneumonia in mice infected with *S. aureus* and treated with ciprofloxacin (where 16.06% of pneumonic areas were obtained in Figure 4c (**** *p* < 0.0001, 95% CI: 8.95, 23.17)) is reduced compared to the infected group without a treatment. In addition, in the analysis of mice lung tissue, infected with *S. aureus* and treated with **7a**, we obtained an average of 5.84% of pneumonic areas (**** *p* < 0.0001, 95% CI: 2.90, 8.77) (Figure 4d), significantly less than the infected mice without treatment. A tendency to a lower pneumonia generation in mice that received **7a**, with respect to those treated with ciprofloxacin, can be seen in Figure 4e.

The results in the described model evidenced the capability of reducing the percentage of pneumonia after administration of **7a** in these conditions, equivalent to that of ciprofloxacin.

## 4. Discussion

The obtained results in the in silico evaluation suggest that **7a** would exhibit antimicrobial activity against *S. aureus* by inhibition of the A subunit of DNA gyrase, which is critical to enzymatic activity [22,32,58]. As we can observe, the most stable binding site predicted for compound **7a** is not the same as the one expected for ciprofloxacin, therefore, we expect activity even in strains showing resistance [59]. This allows us to expect an inhibitory activity by molecule **7a**, which in turn would have advantages over the use of other traditional fluoroquinolones that present resistance due to mutations in their specifical pharmacological target.

The exact mechanism of action of **7a** against *S. aureus* has not yet been fully elucidated, but docking evaluations show possible agreement with that reported in the literature for other fluoroquinolones, regarding QRDR binding differences [60]. The interaction of this compounds with bacterial gyrase leads to bacteriostasis, DNA double helix breaks, chromosomal fragmentation, formation of reactive oxygen species, and finally cell death [61,62]. However, further experimental evidence is still needed in other strains as well as mechanistic experiments.

Regarding in vitro evaluations, (according to CLSI guidelines), areas of inhibition in *S. aureus* > 21 mm show a strong antimicrobial activity against a sensitive strain [44]. It is noteworthy that compound **7a** has both MIC and BMC found at the same concentration. This could be evidence of the slightly higher bactericidal activity of compound **7a**, relative to that of Cpx. We observed a tendency of higher inhibition with 5 μg than with 10 μg, probably due to the lower diffusion of the compound **7a** in the agar by a having higher concentration in the disk [63]. In evaluations performed against wild strains of *S. aureus* where the effect of norfloxacin was evaluated, MICs up to 49.87 μg/mL were obtained [64], while **7a** has a lower MIC possibly due to an increase in its activity because of the addition of the boron ion. Compounds like nemonoxacin have reported MICs below 8 μg/mL [65]. In addition, the MBC of **7a** against *S. aureus* (0. 25 μg/mL) was lower than that of ciprofloxacin (0.5 μg/mL), as well as the MBC of fluoroquinolone derivatives reported by Watanabe et al. with the MBC of 37.5 μg/mL against the same bacteria [66]. And these results are even similar to delafloxacin, a drug used in clinical practice and evaluated by Thabit et al. [67], who report an MIC for *S. aureus* of 0.25 μg/mL. Taken together, the data suggest a comparable or stronger activity of 7a compared to some of the commercially available quinolones.

On the other hand, the antofloxacin, an antibiotic present in the new generation of quinolones, was evaluated against *S. aureus* obtaining MICs of ≤0.25 μg/mL [68]. This compound was also evaluated in a murine model of pneumonia, where it was reported that >10 mg/kg every 12 h was required to obtain bactericidal results against *S. aureus*. In comparison, in our investigation, 80 mg/kg of **7a** was administered, although only once per day. This indicates that **7a** could have a highly effective profile, without the need to administer repeated doses.

Other new generation antibiotics have been evaluated in these murine models of pneumonia. For delafloxacin, to achieve a bactericidal elimination of *K. pneumoniae* in 1-Log with respect to the initial bacterial load, an average of 235 mg/kg/day of delafloxacin was required [69]. This indicates that **7a** could have a highly effective profile and antimicrobial potency with a dosage of 80 mg/kg/day.

The addition of the boron complex was very useful for the quinolonic structure, as evidenced by less pneumonic tissue generation. This may explain the differences in the binding region outside of the QRDR and suggest important differences in the mechanism of action of **7a**. This will merit further experimental and theoretical analysis due to retained antimicrobial activity, even in strains that have developed resistance to QRDR related selection pressure, and where other fluoroquinolones fail as antimicrobials since these also target the QRDR. Although no cytotoxicity data were generated for compound **7a**, the pharmacodynamics and other important pre-clinical data are now essential to take this new molecule beyond preclinical testing and into feasible human trials, given the similar or even improved efficacy compared to well-established and widely used antibiotics. This is in light of the negative in silico assay score and decreased MBC, as has also been observed in other modifications with the same structural position. [61,62,70].

The obtained results highlight compound **7a**’s inhibition of the acute respiratory infection caused by *S. aureus* in mice. Consequently, there are positive expectations concerning its possible future use in therapeutics.

## 5. Conclusions

The molecular docking results revealed the probable inhibition of bacterial DNA gyrase of *S. aureus*, which is consistent with the established mechanism of action for fluoroquinolones, even if the boron complex is added at sites which are usually unmodified in quinolones. Concerning the in vitro and in vivo results, we clearly observed that the boronated fluoroquinolone **7a** is a promising compound with a potent bactericidal effect on *S. aureus* strains. Furthermore, its effectiveness in decreasing the progression of *S. aureus*-induced acute pneumonia was demonstrated in the murine model of intratracheal infection, which allows us to broaden our knowledge of this subfamily of quinolones and expand our research on this and similar compounds.

## Figures and Tables

**Figure 1 cimb-47-00110-f001:**
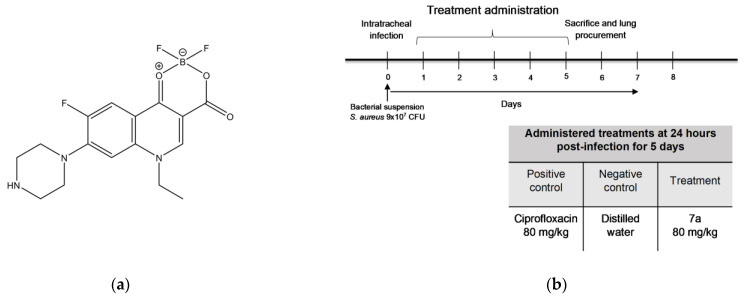
Structure of the **7a** compound and experimental treatment design. (**a**) Chemical structure of the **7a** compound synthesized by our research group. (**b**) Experimental treatment design. Infection was performed at day zero, with 9 × 10^7^ CFU of *S. aureus*. Treatments were administered 24 h post-infection for 5 consecutive days, sacrificing the mice and removing the lungs on day 6.

**Figure 2 cimb-47-00110-f002:**
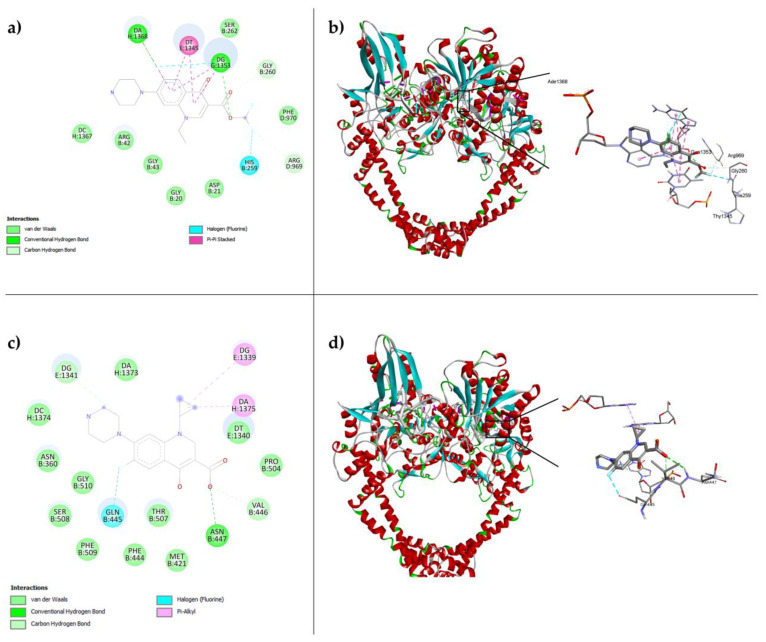
In silico analysis of structure **7a** docking with *S. aureus* gyrase using AutoDock Vina software. (**a**) Interactions predicted for the molecule **7a** with the gyrase amino acids, resulting in the score −10.1 kcal/mol. (**b**) Structure of DNA gyrase (2xct) with the best orientation found for the **7a** molecule was marked with a black rectangle, emphasizing the relevant interactions. (**c**) Interactions predicted for the Cpx with the gyrase, resulting in the score −8.9 kcal/mol. (**d**) Structure of DNA gyrase (2xct) with the best orientation for the Cpx molecule.

**Figure 3 cimb-47-00110-f003:**
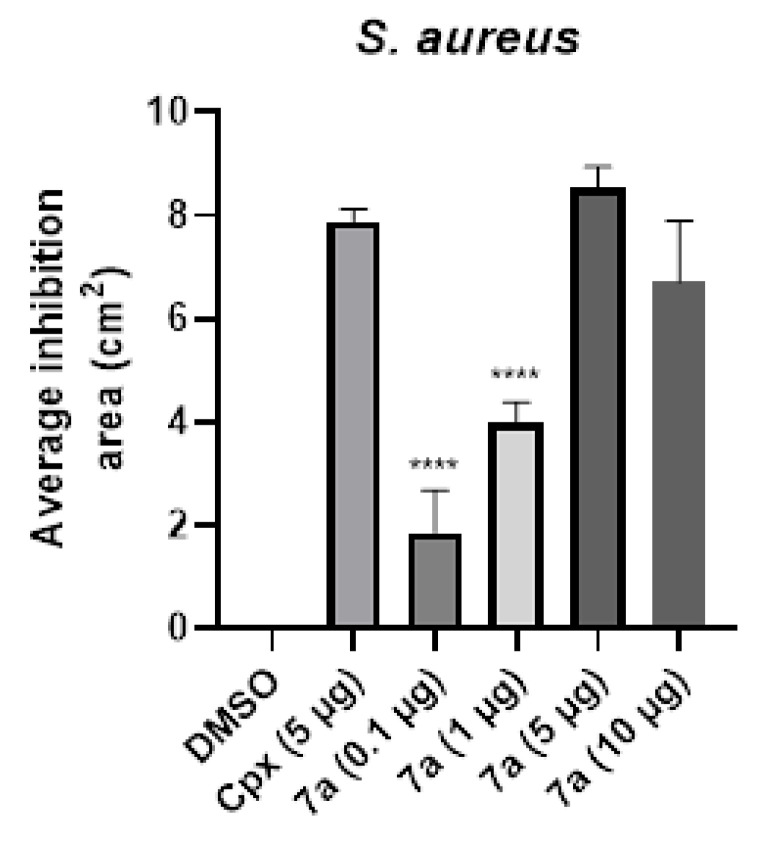
Average inhibition area of **7a** compound against *S. aureus*, Kirby–Bauer evaluation. The mean and standard deviation of the inhibition area of three triplicate experiments of compound **7a** in *S. aureus* culture was plotted. DMSO (vehicle) and 5 μg ciprofloxacin (Cpx) as antibacterial positive control were added. Statistical analysis one-way ANOVA and Tukey’s post hoc test were performed, ns: not significant, **** *p* < 0.0001 significant differences respect to Cpx result.

**Figure 4 cimb-47-00110-f004:**
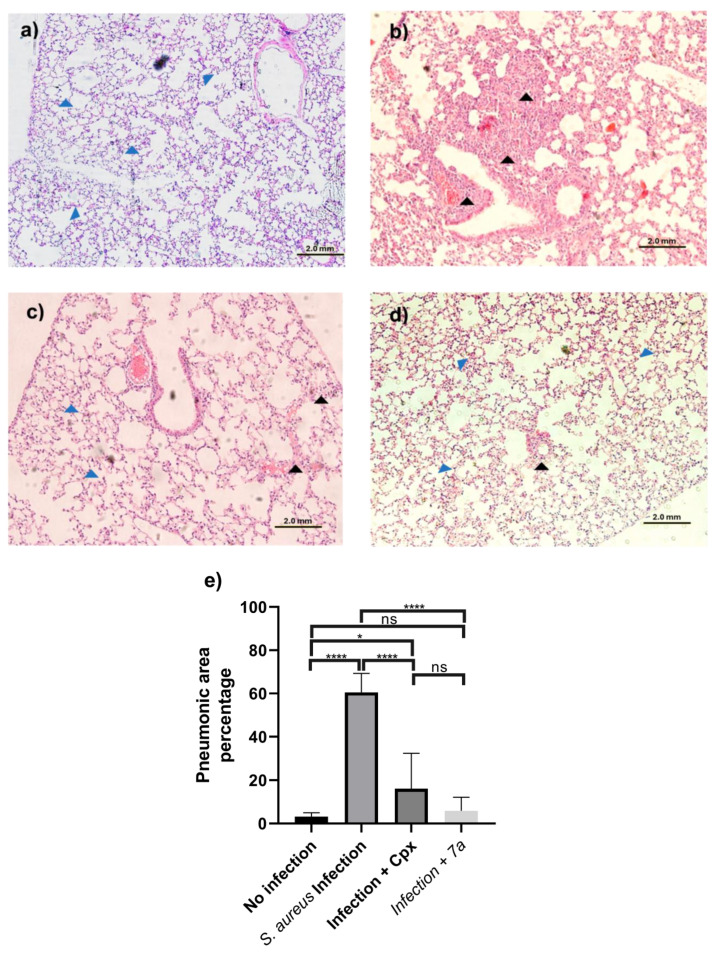
Pneumonic areas in histological lung sections after *S. aureus* infection. Representative images of H&E-stained lung tissue from Balb/c mice n = 4 per experimental group. (**a**) Control mice without infection. (**b**) *S. aureus* infection without treatment; (**c**) *S. aureus* infection with ciprofloxacin treatment. (**d**) *S. aureus* infection with **7a** administration. (**e**) Mean and standard deviation of the percentage of *S. aureus* area determined in experimental groups were plotted. Blue arrowheads point to healthy alveolar tissue, while black arrowheads point to areas of pneumonic injury. Statistical analysis of one-way ANOVA and Tukey post hoc test were performed, where significant differences are expressed as follows: ns = not significant, * *p* < 0.05, **** *p* < 0.0001.

**Table 1 cimb-47-00110-t001:** MIC and MBC results for **7a**.

Bacterial Strain	Compound	MIC (μg/mL)	MBC (μg/mL)
*S. aureus*	Ciprofloxacin	0.25	0.5
	**7a**	0.25	0.25

## Data Availability

The data supporting the findings of this study are available within the article. Further enquiries can be directed to the corresponding author.

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
