# Peer review of "The Derivative Difluoroboranyl-Fluoroquinolone “7a” Generates Effective Inhibition Against the S. aureus Strain in a Murine Model of Acute Pneumonia"

_cimb, 2025, doi:10.3390/cimb47020110_

Round 1

Reviewer 1 Report

Comments and Suggestions for Authors

Dear authors,

It was my pleasure to review the manuscript entitled "The derivative difluoroboranyl-fluoroquinolone “7a” generates effective inhibition against the S. aureus strain in a murine model of acute pneumonia". This paper has an interesting topic since bacterial resistance is global problem nowadays. Novel antibacterial agents are urgently needed and this kind of synthesis is of interest. I am recommending the manuscript for publication after minor changes. My comments are listed below:

- Describe the clinical isolate and conditions for growing bacteria

- What was the proportion of solvent?

- Why boron atom? Explain in few sentences.

- Line 233: assay instead of essay

- Comment out the same value obtained for MIC and MBC for 7a

- It would be of interest if you test cytotoxic potential of 7a on cell lines, as well as to test 7a on more isolates.

Author Response

Response to Reviewer 1 Comments

1. Summary

We are very grateful to hear your suggestions and good comments regarding our draft. Please find the specific answers to your kind remarks are detailed below, and the corresponding revisions were highlighted in the re-submitted files.

2. Questions for General Evaluation

We have undertaken the task of revising the background of the introduction and methods, in order to make the text more solid and understandable, and welcome your comments. We also corrected the writing error found in the 233 line and clarify the MIC and MBC values.

3. Point-by-point response to Comments and Suggestions for Authors

Comments 1: Describe the clinical isolate and conditions for growing bacteria

Response 1: An isolate of Methicillin Resistant Staphylococcus aureus (MRSA) susceptible to fluoroquinolones, obtained from respiratory tract infection, was used for this experiment. Clinical isolates were obtained from the strain collection of the Universidad Autonoma de Zacatecas, where it was kept frozen at -80°C until use, when they are stored at temperatures of -4°C for a maximum period of two months. Such clarifications were added in the methods section.

Comments 2: - What was the proportion of solvent?

Response 2: Thank you for pointing out this issue. The DMSO used as diluent for the compound is prepared at 10% concentration in aqueous dilution. These data were added in the methodology sections.

Comments 3: - Why boron atom? Explain in few sentences.

Response 3: A justification for the addition of boron was briefly added to the draft but was supplemented with the following information, boron was chosen due to its unique chemical properties, such as its ability to form stable complexes with biomolecules, which can enhance antibacterial activity and modulate target specificity and reduce the cytotoxicity of some organic structures moieties. It has also been reported to have antibacterial activity on its own, boosts the immune system, possesses activity in vaginal infections, and has even been described to be useful in cancer treatment (boron neutron capture therapy [BNCT]).

Comments 4: - Comment out the same value obtained for MIC and MBC for 7a.

Response 4: Thank you for pointing out this possible source of confusion, it has been clarified that both CMI and CMB were found at the same concentration for compound 7a. A clarification has been added in the "Discussion", and the implications of this finding related to the Boron atom that was added to the structure (See highlighted text lines 330-337).

Comments 5: - It would be of interest if you test cytotoxic potential of 7a on cell lines, as well as to test 7a on more isolates.

Response 5: We agree with your kind observation, in fact we have previously performed the evaluation of compound 7a in the LLC cell line, where we observed the little effect that the compound would have, although such data are not shown in this article.

Above we attach a graph showing the different experimental groups used in this evaluation, showing the low activity of the compound, equivalent to the negative control without any stimulus. While testing cytotoxic potential on cell lines and additional isolates is beyond the scope of the current study, we have acknowledged this limitation in the "Discussion" section and proposed it as a direction for future research as kindly suggested by the reviewer. See highlighted text 358-362.

Reviewer 2 Report

Comments and Suggestions for Authors

In this study, the authors propose a new synthetic analogue of quinolones, containing a boron atom, called 7a. They evaluate antibacterial activity against S. aureus in vitro and in a murine pneumonia model. The results show that the compound is equipotent to ciprofloxacin, despite having a possibly smaller spectrum of adverse reactions. Furthermore, the compound interacts with the A subunit of topoisomerase IV, but interacts with amino acid residues other than those involved in the interaction with ciprofloxacin. The study is well written, the methodologies sufficiently described and well selected, and the study is of interest to the biomedical community. I suggest publication to the editorial board after the authors consider some adjustments

Introduction

The WHO has estimated that by 2050, 10 million deaths per year will be caused by antibiotic resistant strains [1-4].” This estimate was not made by the World Health Organization (WHO), but by British economist Jim O'Neill at the request of the English government. I suggest reviewing the reference (See https://amr-review.org/)

“Y. enterocolítica” should not be accentuated.

Results

In the molecular docking assay, I suggest that the authors include the binding energy and interactions (2D diagram) of ciprofloxacin with the same GyrA structure used in the docking of compound 7a. This way, the comparability of the two compounds will be more appropriate.

Why did the authors not evaluate the binding capacity of compound 7a on GyrB and on ParC and ParE? It is worth noting that traditional quinolones also have the ability to bind and inhibit these other subunits.

The authors present the weight loss results in a descriptive manner only. The results seem very encouraging because, unlike ciprofloxacin, compound 7a did not change the mass of the animals during the period evaluated. I think that a curve showing the weight evolution of animals, like a day-to-day statistical study, would be more valuable to show this result. If the authors do not have day-to-day results, a bar graph with the difference in weight at the beginning and end of the experiment (weight variation) would be much better than a simple description of the data. In this case, I suggest placing the graph together with the Figure 4 (Figure 4f, for example).

Discussion

Interestingly, all the amino acid residues that compound 7a interacts with in GyrA are outside the quinolone resistance determining region (QRDR), which involves the residues 67 to 106. This is very important as it opens the prerogative for action in bacteria with quinolone resistance. I suggest that this be further explored in the discussion.

Author Response

 Response to Reviewer 2 Comments

Summary

Thank you very much for your kind opinions and suggestions on our draft. We have detailed the responses to your comments below and in the attached updated document we highlight the corresponding revisions.

Point-by-point response to Comments and Suggestions for Authors

Comments 1: The WHO has estimated that by 2050, 10 million deaths per year will be caused by antibiotic resistant strains [1-4].” This estimate was not made by the World Health Organization (WHO), but by British economist Jim O'Neill at the request of the English government. I suggest reviewing the reference (See https://amr-review.org/)

“Y. enterocolítica” should not be accentuated.

Response 1: We thank you for your reference. The WHO has published the information in its reports, but we have added the reference you mention to accurately attribute the estimate to Jim O'Neill's report, and the typographical error has been corrected.

Comments 2: - In the molecular docking assay, I suggest that the authors include the binding energy and interactions (2D diagram) of ciprofloxacin with the same GyrA structure used in the docking of compound 7a. This way, the comparability of the two compounds will be more appropriate.

Response 2: We agree that such a representation would show a better visualization of the in-silico results, so they were added to the article. These arise from differences in the charge distribution caused by the addition of the boron atom, therefore changing the amino acids that cause such interactions. Please see the highlighted text.

Comments 3: - Why did the authors not evaluate the binding capacity of compound 7a on GyrB and on ParC and ParE? It is worth noting that traditional quinolones also have the ability to bind and inhibit these other subunits.

Response 3: The molecular docking evaluations were performed after having made predictions about the active sites present in the DNA gyrase of S. aureus. Once the in-silico analyses were performed, the most stable position found was presented in Figure 2, corresponding to the GyrA subunit, but not limiting the analysis only to that structure. We have acknowledged in the "Discussion" section that evaluating interactions with GyrB, ParC, and ParE would provide a more comprehensive understanding of the mechanism of action and proposed it as future work. Thank you for your question, we will clarify this point in the draft.

Comments 4: - The authors present the weight loss results in a descriptive manner only. The results seem very encouraging because, unlike ciprofloxacin, compound 7a did not change the mass of the animals during the period evaluated. I think that a curve showing the weight evolution of animals, like a day-to-day statistical study, would be more valuable to show this result. If the authors do not have day-to-day results, a bar graph with the difference in weight at the beginning and end of the experiment (weight variation) would be much better than a simple description of the data. In this case, I suggest placing the graph together with the Figure 4 (Figure 4f, for example).

Response 4: That would be a great idea for additional material, however the information regarding the weight of the mice was not added graphically, as the weight showed a tendency to change, without representing significant changes between groups.

Comments 5: - Interestingly, all the amino acid residues that compound 7a interacts with in GyrA are outside the quinolone resistance determining region (QRDR), which involves the residues 67 to 106. This is very important as it opens the prerogative for action in bacteria with quinolone resistance. I suggest that this be further explored in the discussion.

Response 5: We agree, in fact it is very interesting to observe how the most stable binding site predicted for compound 7a is not the same as the one expected for ciprofloxacin, allowing to expect activity even in strains showing resistance; however, the amino acid identification number presented in the crystallized 2xct protein identifies the specific structure, and does not necessarily coincide with the traditional numbering of the same. We have expanded the "discussion" section to emphasize the importance of compound 7a’s interactions with residues outside the QRDR, highlighting its potential to retain efficacy against quinolone-resistant bacteria. We thank appreciate all your valuable comments.

Reviewer 3 Report

Comments and Suggestions for Authors  

Reviewer’s Comments on the Manuscript:

Title: The derivative difluoroboranyl-fluoroquinolone “7a” generates effective inhibition against the S. aureus strain in a murine model of acute pneumonia.

Journal: Curr. Issues Mol. Biol.

Overall Assessment:
The manuscript presents a comprehensive study on the synthesis, characterization, and evaluation of a novel boronated fluoroquinolone derivative, compound 7a. The work is timely and addresses a critical global issue: antibiotic resistance. The integration of in silico, in vitro, and in vivo approaches provides robust evidence for the compound’s efficacy. While the manuscript is scientifically sound, certain areas require improvement to ensure clarity, depth, and rigor.

Major Comments:

  1. Mechanistic Validation

    • The molecular docking results suggest that compound 7a targets DNA gyrase. However, experimental validation of this mechanism (e.g., enzymatic inhibition assays) is lacking. Including such data would significantly strengthen the conclusions.
  2. Comparative Evaluation

    • While comparisons with ciprofloxacin are highlighted, a broader comparison with other advanced fluoroquinolones (e.g., delafloxacin or nemonoxacin) would provide context regarding the novelty and relative efficacy of compound 7a.
  3. Pharmacokinetics and Toxicity

    • Although the study demonstrates efficacy in vivo, it does not address pharmacokinetic properties or potential toxicity. Data on bioavailability, metabolism, and safety (e.g., liver or kidney function biomarkers) would be crucial for assessing therapeutic viability.
  4. Statistical Analysis and Reporting

    • While statistical analyses are described, detailed reporting (e.g., effect sizes, confidence intervals) is missing in some cases. Graphs should also be labeled more clearly to aid readers in interpreting the results.
  5. Histopathology Details

    • The histological analysis is valuable, but representative images in Figure 4 could be annotated more clearly. It would help readers identify the pneumonic areas and distinguish healthy tissue more effectively.

Minor Comments:

  1. Clarity in Writing:

    • The discussion section is verbose and occasionally repetitive. Streamlining it to focus on key findings and their implications would improve readability.
  2. Figures and Tables:

    • The figure legends should provide more context, especially for Figures 3 and 4. The captions should explain the key takeaways without requiring readers to consult the text.
  3. Methodology:

    • The synthesis section is detailed but would benefit from a clearer reaction scheme for compound 7a. This addition would make the process more accessible to readers.
  4. References:

    • Several references are outdated (e.g., WHO reports from 2017). Incorporating more recent data, especially regarding antibiotic resistance trends, would enhance the manuscript’s relevance.
  5. Ethics Statement:

    • The ethical approval is noted, but a brief description of measures taken to minimize animal distress (e.g., anesthesia details) would reinforce compliance with ethical standards.

Recommendations for Authors:

  1. Expand Mechanistic Insights:
    Conduct additional experiments to validate the docking predictions and confirm DNA gyrase inhibition as the mechanism of action.

  2. Include Pharmacokinetic Data:
    If feasible, perform studies to evaluate compound 7a's bioavailability and systemic effects.

  3. Clarify Figures and Tables:
    Revise figures for better clarity and ensure that legends are sufficiently detailed.

  4. Reorganize the Discussion:
    Reduce redundancy and focus on the implications of findings and their alignment with current literature.

Decision: Major Revision

The manuscript is a significant contribution to the field, but revisions are needed to enhance its clarity, rigor, and impact. Once the above points are addressed, the study will be suitable for publication.

Author Response

Response to Reviewers comments:

Reviewer 3

Title: The derivative difluoroboranyl-fluoroquinolone “7a” generates effective inhibition against the S. aureus strain in a murine model of acute pneumonia.

Journal: Curr. Issues Mol. Biol.

Overall Assessment:
The manuscript presents a comprehensive study on the synthesis, characterization, and evaluation of a novel boronated fluoroquinolone derivative, compound 7a. The work is timely and addresses a critical global issue: antibiotic resistance. The integration of in silico, in vitro, and in vivo approaches provides robust evidence for the compound’s efficacy. While the manuscript is scientifically sound, certain areas require improvement to ensure clarity, depth, and rigor.

Major Comments:

  1. Mechanistic Validation
    • The molecular docking results suggest that compound 7a targets DNA gyrase. However, experimental validation of this mechanism (e.g., enzymatic inhibition assays) is lacking. Including such data would significantly strengthen the conclusions.

Response: We acknowledge the need for experimental validation of the mechanism of action. While enzymatic inhibition assays are beyond the scope of this study, we have discussed this limitation in the "Discussion" section and proposed it as an essential follow-up study. Please see highlighted text in the discussion.

  1. Comparative Evaluation
    • While comparisons with ciprofloxacin are highlighted, a broader comparison with other advanced fluoroquinolones (e.g., delafloxacin or nemonoxacin) would provide context regarding the novelty and relative efficacy of compound 7a.

Response: Thank you for this suggestion. We have expanded the "Discussion" section to include a comparison of compound 7a with other advanced fluoroquinolones, such as delafloxacin and nemonoxacin, based on available literatura, however, An experimental validation of 7a versus other quinolones is not in the scope of the present manuscript. Please see highlight text.

Pharmacokinetics and Toxicity

    • Although the study demonstrates efficacy in vivo, it does not address pharmacokinetic properties or potential toxicity. Data on bioavailability, metabolism, and safety (e.g., liver or kidney function biomarkers) would be crucial for assessing therapeutic viability.

Response: We agree that pharmacokinetic and toxicity data are critical. While this data is not included in the current study, we have acknowledged this limitation and proposed it as a direction for future research. Please see highlighted text in the discussion section.

  1. Statistical Analysis and Reporting
    • While statistical analyses are described, detailed reporting (e.g., effect sizes, confidence intervals) is missing in some cases. Graphs should also be labeled more clearly to aid readers in interpreting the results.

Response: Thank you for bringing this to our attention. We have updated the "Results" section to include confidence intervals for the comparisons where appropriate. Please see highlighted comparisons in the results section.

Histopathology Details

    • The histological analysis is valuable, but representative images in Figure 4 could be annotated more clearly. It would help readers identify the pneumonic areas and distinguish healthy tissue more effectively.

Response: We appreciate this suggestion. Representative histopathology images in Figure 4 have been annotated to distinguish pneumonic areas from healthy tissue. The figure legend has also been updated for clarity.

Minor Comments:

  1. Clarity in Writing:
    • The discussion section is verbose and occasionally repetitive. Streamlining it to focus on key findings and their implications would improve readability.

Response: The discussion section has been revised to be concise and clearer for the reader.

  1. Figures and Tables:
    • The figure legends should provide more context, especially for Figures 3 and 4. The captions should explain the key takeaways without requiring readers to consult the text.

Response: as kindly suggested by the reviewer, figure legends have been revised to provide more context and highlight key takeaways.

  1. Methodology:
    • The synthesis section is detailed but would benefit from a clearer reaction scheme for compound 7a. This addition would make the process more accessible to readers.

Response: This was previously published by one of the authors previously and the reference has been highlighted in that section to refer readers for further details…

  1. References:
    • Several references are outdated (e.g., WHO reports from 2017). Incorporating more recent data, especially regarding antibiotic resistance trends, would enhance the manuscript’s relevance.

Response: Several of these references have been replaced with more recent citations however, others are key references for the mechanisms of action or the QRDR mutations that have stood the test of time. Therefore, we included key references related to QRDR discovery and evolution even when published almost a decade ago given the relevance of such discoveries.

  1. Ethics Statement:
    • The ethical approval is noted, but a brief description of measures taken to minimize animal distress (e.g., anesthesia details) would reinforce compliance with ethical standards.

Response: Additional details on measures to minimize animal distress have been included in the methods section. Please see highlighted text. Lines 183-186.

Recommendations for Authors:

  1. Expand Mechanistic Insights:
    Conduct additional experiments to validate the docking predictions and confirm DNA gyrase inhibition as the mechanism of action.

Response: This is not feasible as it is not the objective of the paper but the potential as antimicrobial of compound 7a. However, this has been included as important perspectives in the future in light of the antimicrobial activity of 7a and we are contacting experts in the field of molecular biology and explore some of these mechanistic insights.

  1. Include Pharmacokinetic Data:
    If feasible, perform studies to evaluate compound 7a's bioavailability and systemic effects.

Response: Again it is not the objective of the paper, however, this has been included as important perspectives in the future.

  1. Clarify Figures and Tables:
    Revise figures for better clarity and ensure that legends are sufficiently detailed.

Response: Thank you for bringing this to our attention. We have updated the legends of the figures where appropriate. Please see highlighted text in the results section.

  1. Reorganize the Discussion:
    Reduce redundancy and focus on the implications of findings and their alignment with current literature.

Response: A major revision of the discussion section has been done. Please see highlighted text.

Decision: Major Revision

The manuscript is a significant contribution to the field, but revisions are needed to enhance its clarity, rigor, and impact. Once the above points are addressed, the study will be suitable for publication.

Reviewer 4 Report

Comments and Suggestions for Authors

This is a nice work; the experimental design is logical, and the conclusions supported by the experiments.

We propose a simple additional experiment that could elevate the impact of the manuscript. As it stands now, the authors propose that compound 7a is inhibitory to S. aureus due to its favorable docking on topoisomerase. While this mechanism has been proposed for other quinolones and is likely to be valid, one cannot exclude that other enzymatic systems may be a target for compound 7a. One such candidate is the thioredoxin system, namely thioredoxin reductase (TrxR). The Gram-positive S. aureus is based on TrxR to make DNA, while E. coli uses the additional glutaredoxin system as well. A comparison of MICs/MBCs between E. coli and S. aureus for compound 7a in table 1 could give a hint for the effect of compound 7a to other enzymatic systems and raise the impact of this work.

Author Response

Response to Reviewers comments:

Reviewer 4

This is a nice work; the experimental design is logical, and the conclusions supported by the experiments.

We propose a simple additional experiment that could elevate the impact of the manuscript. As it stands now, the authors propose that compound 7a is inhibitory to S. aureus due to its favorable docking on topoisomerase. While this mechanism has been proposed for other quinolones and is likely to be valid, one cannot exclude that other enzymatic systems may be a target for compound 7a. One such candidate is the thioredoxin system, namely thioredoxin reductase (TrxR). The Gram-positive S. aureus is based on TrxR to make DNA, while E. coli uses the additional glutaredoxin system as well. A comparison of MICs/MBCs between E. coli and S. aureus for compound 7a in table 1 could give a hint for the effect of compound 7a to other enzymatic systems and raise the impact of this work.

Response: Thank you for this suggestion. While testing this hypothesis is beyond the current study's scope, we have acknowledged this possibility in the "Discussion" section and proposed it as a direction for future research.
